# Control of acute myeloid leukemia and generation of immune memory *in vivo* using AMV564, a bivalent bispecific CD33 x CD3 T cell engager

**Linda G. Eissenberg[1], Julie K. Ritchey[1], Michael P. Rettig[1], Dilan A. Patel [1], Kiran Vij[2], Feng Gao[3], Victoria Smith[4], Tae H. Han[4], John F. DiPersio[1] ***

**1** Department of Internal Medicine, Division of Oncology, Washington University, St. Louis, Missouri, United States of America, **2** Department of Pathology, Washington University, St. Louis, Missouri, United States of America, **3** Department of Surgery, Division of Public Health Sciences, Washington University, St. Louis, Missouri, United States of America, **4** Amphivena Therapeutics, Inc., San Francisco, California, United States of America

\* jdipersi@wustl.edu

**Data Availability Statement:** We have uploaded raw data files to the Harvard Dataverse. The link is

## Abstract

Off-the-shelf immunotherapeutics that suppress tumor growth and provide durable protection against relapse could enhance cancer treatment. We report preclinical studies on a CD33 x CD3 bivalent bispecific diabody, AMV564, that not only suppresses tumor growth, but also facilitates memory responses in a mouse model of acute myelogenous leukemia (AML). Mechanistically, a single 5-day treatment with AMV564 seems to reduce tumor burden by redirection of T cells, providing a time window for allogeneic or other T cells that innately recognize tumor antigens to become activated and proliferate. When the concentration of bispecific becomes negligible, the effector: target ratio has also shifted, and these activated T cells mediate long-term tumor control. To test the efficacy of AMV564 *in vivo*, we generated a CD33$^+$ MOLM13$^{CG}$ bioluminescent human cell line and optimized conditions needed to control these cells for 62 days *in vivo* in NSG mice. Of note, not only did MOLM13$^{CG}$ become undetectable by bioluminescence imaging in response to infusion of human T cells plus AMV564, but also NSG mice that had cleared the tumor also resisted rechallenge with MOLM13$^{CG}$ in spite of no additional AMV564 treatment. In these mice, we identified effector and effector memory human CD4$^+$ and CD8$^+$ T cells in the peripheral blood immediately prior to rechallenge that expanded significantly during the subsequent 18 days. In addition to the anti-tumor effects of AMV564 on the clearance of MOLM13$^{CG}$ cells *in vivo*, similar effects were seen when primary CD33$^+$ human AML cells were engrafted in NSG mice even when the human T cells made up only 2% of the peripheral blood cells and AML cells made up 98%. These studies suggest that AMV564 is a novel and effective bispecific diabody for the targeting of CD33$^+$ AML that may provide long-term survival advantages in the clinic.

posted below. https://doi.org/10.7910/DVN/QKEPPK.

**Funding:** The work was supported by National Institutes of Health (NIH)/National Cancer Institute (NCI): R35 CA210084 (JFD), NIH/NCI: P50 CA171963 NCI (JFD) and NIH/NCI R50 CA211466-06 (MPR). The funders had no role in study design, data collection and analysis, decision to publish, or preparation of the manuscript.

**Competing interests:** The authors have declared that no competing interests exist.

## Introduction

Bispecifics (both small proteins and antibodies) facilitate immune synapse between CD3 epsilon on T cells and selected differential expressed antigens on tumors. Several bispecifics have been developed to target CD33 [1], an antigen differentially expressed in ~80–90% of all AML patients.

AMV564 is a tetravalent tandem diabody [2] with two high-avidity recognition sites for CD33 and two for the invariant CD3-epsilon chain found in the T cell receptor. Since it is smaller than whole immunoglobulins, it is expected to penetrate better into tissues, yet it is large enough to be spared from first pass clearance through the kidneys, in contrast to bispecific T cell engagers (BiTEs). Structurally related molecules have a half-life between 18–23 hours in mice [3], so AMV564 could conceivably be administered via daily injections rather than necessitating continuous infusion by pumps over several weeks, as is needed for BiTEs [1].

In this study, we aimed to determine the efficacy and durability of response to treatment with AMV564. We show that AMV564, in conjuction with human T cells, can eliminate AML cells (both luciferase-expressing cell lines and primary AML cells) *in vivo* from NSG mice. Using serial bioluminescent imaging (BLI), we followed tumor cells in individual immunodeficient mice throughout these experiments. This allowed for optimization of the treatment schedule and resulted in substantial and extended tumor control. Unexpectedly, we found that a single 5-day treatment cycle induced protective immunity that resulted in resistance to tumor rechallenge. We hypothesize that this is due to an early reduction in tumor burden, during which time allogeneic T cells, and perhaps T cells with innate recognition of antigens on the tumor, become activated and capable of sustained tumor control.

## Methods

### Reagents

AMV564 refers to either the research grade (T564) or clinical grade version of Amphivena's tetravalent bispecific huCD33 x huCD3 diabody [2, 4]. It binds both human CD33 and human CD3ε with strong avidity and specificity [2]. AMV564 is being evaluated in clinical trials for patients with relapsed and refractory AML (NCT03144245 [4, 5]), MDS (NCT03516591 [6]), and solid tumors (NCT0412842 [7]). T151 is a similarly constructed bispecific targeting human serum albumin.

### Ethics statement

Written informed consent for use of collected cells was provided by patient UPN789148, who was enrolled in a tissue banking protocol that was approved by the Washington University Human Studies Committee (HSC#01–1014) in accordance with the Declaration of Helsinki. This patient had AML subtype M5B.

### Cells and cell lines

The human cell lines MOLM13 and Jurkat were obtained from Deutsche Sammlung von Mikroorganismen und Zellkulturen, Germany. Ramos and KG1 were obtained from the American Type Culture Collection (ATCC). All were cultured in RPMI-1640 and 10% fetal bovine serum (FBS, Atlanta Biologicals), 1% GlutaMax (Life Technologies, Invitrogen), 50 μM 2-mercaptoethanol (Sigma-Aldrich), and 1% penicillin-streptomycin (Cellgro) at 37°C with 5% $CO_2$. Cells were transduced with click beetle red luciferase and enhanced green fluorescent protein using EF1α$^{CBR-GFP}$ lentivirus [8]. Cells were purified by fluorescence-activated cell sorting and cloned to establish the "CG" versions of the cell lines. T cells were isolated from purchased

leukopacks from healthy donors (Miltenyi Biotec Inc, Auburn, CA). Miltenyi Pan T selection Kits and an AutoMacs Pro (Miltenyi Biotec) were used to negatively enrich for T cells.

For *in vitro* and *in vivo* experiments using tumor cell lines, negatively selected and purified CD3[+] healthy donor T cells were used. For *in vitro* analysis of T cell activation and tumor cell killing, $2x10^4$ MOLM13[CG] (CD33[+]), KG1[CG] (CD33[+]), or Ramos[CG] cells (CD33[-]) were co-incubated at 37˚C in 200 ml RPMI-1640 medium in a microtiter plate ± AMV564 at the designated concentrations and a sufficient number of T cells to provide the indicated effector: target ratios. At the specified times, the cells were washed and processed for flow cytometry to detect the markers indicated in the figure legends.

## Animal studies

Six to seven week old immunodeficient male NOD.Cg-Prkdc[scid] Il2rg[tm1Wjl]/SzJ (NSG) mice were obtained from The Jackson Laboratory (Bar Harbor, ME) and housed in a pathogen-free environment and maintained on ad libitum water and chow (LabDiet 5053; Lab Supply, Fort Worth, TX), a 12 h light/12 h dark cycle, a temperature range of 68–74˚F with 40–60% humidity, and pulped virgin cotton nesting squares for environmental enrichment. Mice were acclimated for 3–7 days before use in an experiment. Animal protocols complied with the regulations of Washington University School of Medicine Animal Studies Committee, (IACUC protocol #20–0467) and were conducted in accordance with the NIH guidelines for housing and care. All efforts were made to minimize suffering, including provision of Nutragel (Bio-Serv, Flemington, New Jersey) for slight mobility issues. Facility staff and laboratory scientists conducting the experiments undergo training via Washington University's Department of Comparative Medicine using training modules that include: "Animal use regulations," "Rats and mice," "Animal Biohazard Safety," and "Rodent Barrier Orientation." In addition, they receive hands-on training from experienced labmates.

Animal health and behavior were monitored twice daily. Mice that appeared moribund were humanely euthanized by carbon dioxide overdose within 24 h when a total score of 6 was reached using predefined and IACUC approved criteria for moribundity (S1 Table). In addition, a weight loss of ≥30%, hind limb paralysis, or a tumor mass of ≥ 2 cm, regardless of total score, necessitated euthanasia. In this model most mice are asymptomatic until a day or two before needing to be sacrificed. Occasional mice show no symptoms before their death and are found dead. In the studies reported here, all such mice were then found to have either a tumor mass, BLI evidence of tumor expansion, or both.

Upon arrival, mice are randomized into cages of five animals by facility staff who have no knowledge of the experiment to be done, and the cages are randomly placed on a rack. The first cage on the rack is used for the first group, and so on. The groups are arranged in the same order for each experiment. The number of mice/group reflects our attempt to minimize the number of mice used, while still providing statistically meaningful comparisons between groups based on our previous experience with the MOLM13[CG]/NSG mouse model and the anticipated variability between individuals in each group. Untreated mice typically succumb within 2–3 days of each other, by ~ day 26. In all cases, the primary outcome measure was the number of surviving mice at a predefined time point. The number of animals in each experiment, their causes of death, and the duration of experiments can be found in S2 Table. Individuals involved in conducting, assessing outcome, and analyzing the data were aware of the allocations. The statistician was unaware of the hypotheses being tested.

Mice were irradiated (250 cGy, JL Shepherd and Associates, San Fernando, California) 3 days prior to i.v. injection with $5 x 10^6$ AML patient peripheral blood mononuclear cells (PBMC). A differential cell count on the peripheral blood from patient UPN789148

demonstrateded 98% myeloblasts and 2% T cells. No conditioning was done before i.v. injection of 3.3 x $10^3$ MOLM13$^{CG}$ cells. For the latter studies, 2–8 x $10^6$ T cells prepared as above were injected i.v. once between days 3 and 17 as indicated in each study. Bispecific antibody was likewise administered i.v.at the time indicated in each figure. Flow cytometry was used to follow the tumor (primary AML or MOLM13$^{CG}$) or human T cell and human T cell subsets in the peripheral blood and/or harvested organs. BLI was used to follow the kinetics of MOLM13$^{CG}$ expansion or elimination in individual NSG mice. Suspected visible tumor masses in the epididymis, testes, or small intestines, as well as random skin samples from the backs of mice, were histologically reviewed using paraffin-embedded tissues which were sectioned and H and E stained by the Division of Comparative Medicine at Washington University.

### Flow cytometry

S3 Table lists the panel of markers and antibodies used for flow cytometry analyses in each experiment. A Gallios flow cytometer (Beckman Coulter Life Sciences, Indianapolis, Indiana) was used for all experiments except the rechallenge experiment, in which we analyzed T cell subsets [9] using a ZE5 Attune NxT flow cytometer (Bio-Rad, Hercules, California). Collected data were analyzed using Flow Jo V10 (TreeStar) software. Accucount beads (Spherotech, Inc., Lake Forest, Illinois) were used to standardize cell counts. Murine cell marker: muCD45. Tumor cell markers: GFP, CD33, CD123. Human T cell markers: CD3, CD4, CD8, CD25, CD45RA, CD56. T cell subset phenotypic markers: naïve cells (CD197$^+$CD45RA$^+$), central memory cells (CD197$^+$CD45RA$^{neg}$), effector cells (CD197$^{neg}$CD45RA$^{neg}$), effector memory cells (CD197$^{neg}$CD45RA$^+$). Viability: 7-aminoactinomycin-D (7-AAD, BD Biosciences, Franklin Lakes, New Jersey) or Live/Dead stain (Fisher Scientific, Pittsburgh, Pennsylvania).

The quantity of T cells (CD3 marker) or tumor cells (GFP) in a tissue or organ was calculated as follows: (Total tissue or organ volume [microliters] /sample volume read on cytometer [microliters]) X number of events counted in the sample.

### Statistical analyses

Mouse survival was described using the Kaplan-Meier product limit method and compared by the Wilcoxon test. All other data were summarized using means and standard deviations. For tumor burden followed by bioluminescence imaging, linear mixed models were used to describe the change over time, followed by post-hoc multiple comparisons for mean levels at specific times or average over time changes between groups of interest. The number of cells per organ was evaluated using one-way ANOVA. The normality of data was assessed graphically using residuals and logarithm or square-root data transformation was performed as appropriate to better satisfy the normality and homoscedasticity assumptions. All analyses were two-sided and significance was set at a p-value of 0.05. The statistical analyses were performed using SAS 9.4 (SAS Institutes, Cary, NC). The assumption of normality distribution and homoscedasticity (i.e., equal variance) was assessed graphically using residuals. Data transformation using square-root (for data containing values less than 1) or logarithm was performed as appropriate to better satisfy the assumption.

## Results

### AMV564 induces T cell functionality at low effector-to-target ratios

For a bispecific to have clinical utility, it needs to activate circulating T cells even when they are vastly outnumbered by tumor cells, a common scenario in AML patients. We extended the work of others on the bispecific bivalent AMV564 (Fig 1A, [2]) to determine the lowest

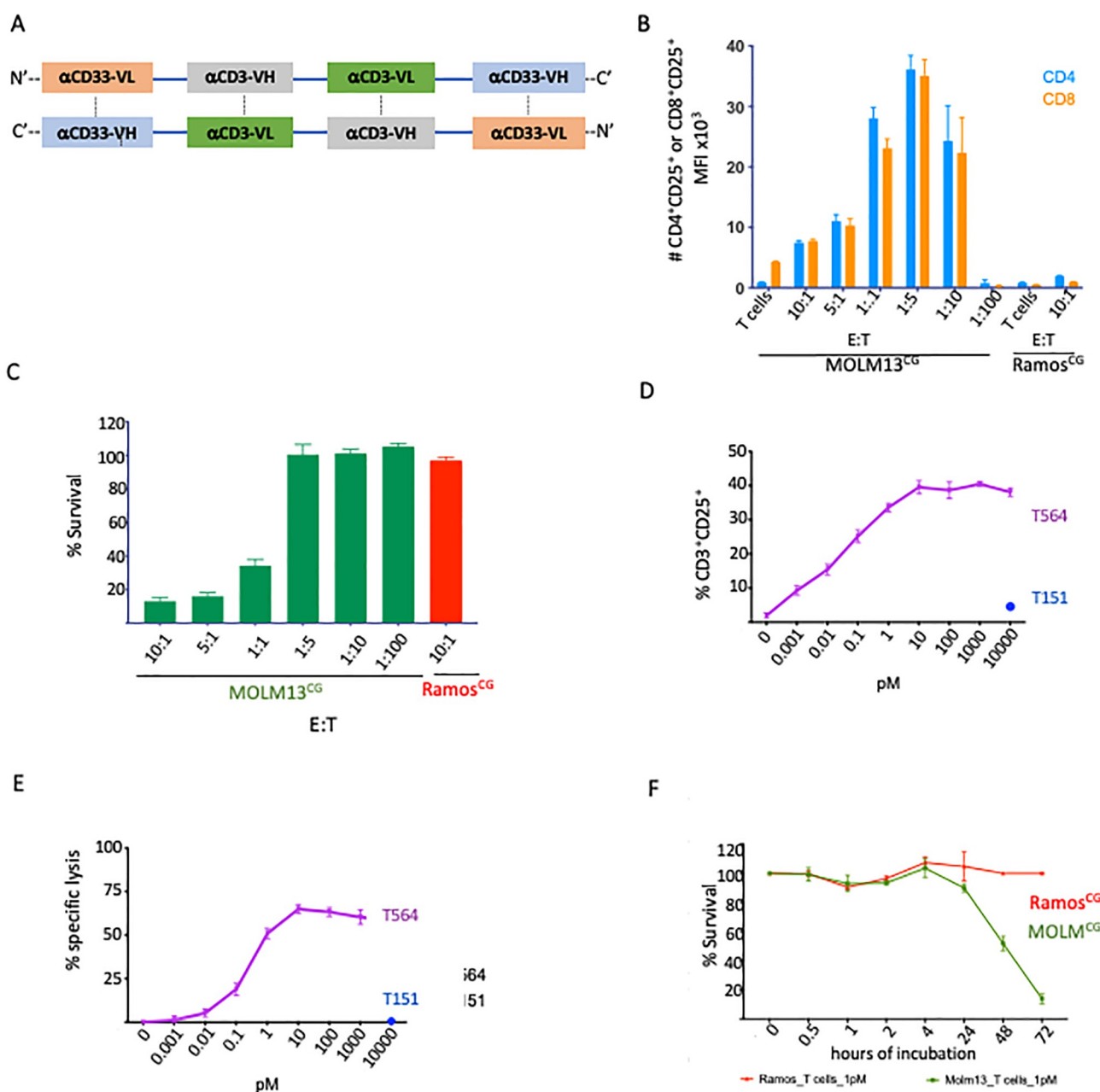

**Fig 1. Effect of E:T ratio, AMV564 concentration, and time on T cell function.** (A) AMV564 is a diabody composed of two identical proteins encoding Vh and Vl sequences for anti-CD33 and anti-CD3 that non-covalently bind together to form a bivalent bispecific [2]. (B) Negatively selected CD3[+] T cells were co-incubated with 2 x 10^4 MOLM13[CG] cells (CD33[+]) or Ramos[CG] cells (CD33[neg]) at various E:T ratios along with 1pM AMV564. Flow cytometry was used to detect activation of human T cells, evidenced as an increase in the number of CD4[+]CD25[+] and CD8[+]CD25[+] cells and (C) killing of target cells at 72 h. Flow cytometry markers = 7AAD (viability), GFP, human CD3, CD4, CD8, CD25. (D) T564 (the reagent grade version of the clinical compound, AMV564; purple) were added at various concentrations to T cells and KG1[CG] cells (CD33[+]; E:T = 1:5) and resulted in activation of the T cells and (E) lysis of the target cells. T151 (blue), a similar bispecific targeting human serum albumin, was tested only at 10,000pM. (F) Kinetics of MOLM13[CG] cell killing over 72 h using 1 pM AMV564 and an E:T ratio of 1:1. N = 3 replicates, 1 experiment in B-F.

effector: target (E:T) ratio needed to activate healthy donor T cells and mediate cytolysis against the AML cell line, MOLM13$^{CG}$ *in vitro*. Using 1 pM AMV564, even an E: T ratio as low as 1:10, induced activation of T cells within 72 h (Fig 1B). At the same time point, cytolytic activity was clearly evident at an E:T ratio of 1:1 at 72 h (Fig 1C). In studying the effect of drug concentration at 24 h and using another CD33$^+$ AML cell line, KG1$^{CG}$, we held the E:T ratio at 1:5. Under these conditions, AMV564 mediated near maximal T cell activation (Fig 1D) and cytolysis (Fig 1E) at just 1 pM AMV564. In contrast, 10,000 pM T151, a structurally related bispecific targeting human serum albumin, had no effect as expected. In a study of kinetics at an E:T of 1:1 and using 1 pM drug, cytolysis became detectable at 24 h and was nearly complete by 72 h (Fig 1F). These experiments indicated that AMV564 rapidly induces T cell activation and anti-tumor killing at low concentrations and E:T ratios and encouraged us to progress to *in vivo* studies.

## AMV564 + human T cells kill MOLM13$^{CG}$ in NSG mice *in vivo*

Since AML is a liquid tumor, we studied AMV564 efficacy using intraveneously injected tumor cells to better model a patient situation than the solid tumor models used by others [2]. In this model, we inject 3.3x10$^3$ MOLM13$^{CG}$ cells intravenously (i.v.) into NSG mice on day 0 and allow them to engraft. Growth of this aggressive AML cell line is detectable by bioluminescence imaging as early as 3 days later. The tumor cells quickly home to the bone marrow, just as in patients, and they proliferate there until the marrow becomes packed. Tumor cells do not enter the peripheral circulation until shortly before the mice die. Untreated mice become moribund and require sacrifice around day 25. Using this established tumor model, we can follow each mouse individually throughout the experiment.

### One treatment cycle is as effective as two at increasing mouse survival

We first tested whether AMV564 could enhance tumor control and prolong mouse survival, and if so, whether two treatment cycles were more effective than one (schema, Fig 2A). We administered a single dose of 2 x10$^6$ purified healthy donor human T cells to mice seven days after injecting MOLM13$^{CG}$ tumor cells. Test groups also received one or two 4-day cycles of 5 mcg i.v. AMV564 daily. Control groups were left untreated, administered 2 cycles of AMV564 but no human T cells, or were only administered T cells. The first dose of AMV564 was administered on the same day that the T cells were injected.

Treating mice with either one or two cycles of AMV564 for four days each modestly increased mouse survival from ~23 days to ~36 days post tumor injection (Fig 2B). BLI indicated that mean tumor growth in the treated groups was only briefly suppressed, then resumed at nearly the same rate as in untreated mice (Fig 2C). In fact, by tracking the bioluminescence in individual mice (S1 Fig), we learned that only a few of the mice in each group actually showed any early decrease in BLI signal on day 14. Nonetheless, consistent with the overall improvement in mouse survival, while there were many circulating MOLM13$^{CG}$ cells detected in control mice on day 23, none were detected in 9 out of 10 mice treated with two cycles of AMV564 (S2 Fig). We did not assess the blood of mice treated with only a single AMV564 cycle. However, since the BLI signal and survival were similar between the two groups, we opted to use only a single AMV564 cycle in subsequent experiments.

### Mouse survival is more dependent upon T cell number than on AMV564 dose

In the previous experiment, we noted that some mice had better tumor control (by BLI) than others and that the mean rate of tumor proliferation was initially suppressed, then continued

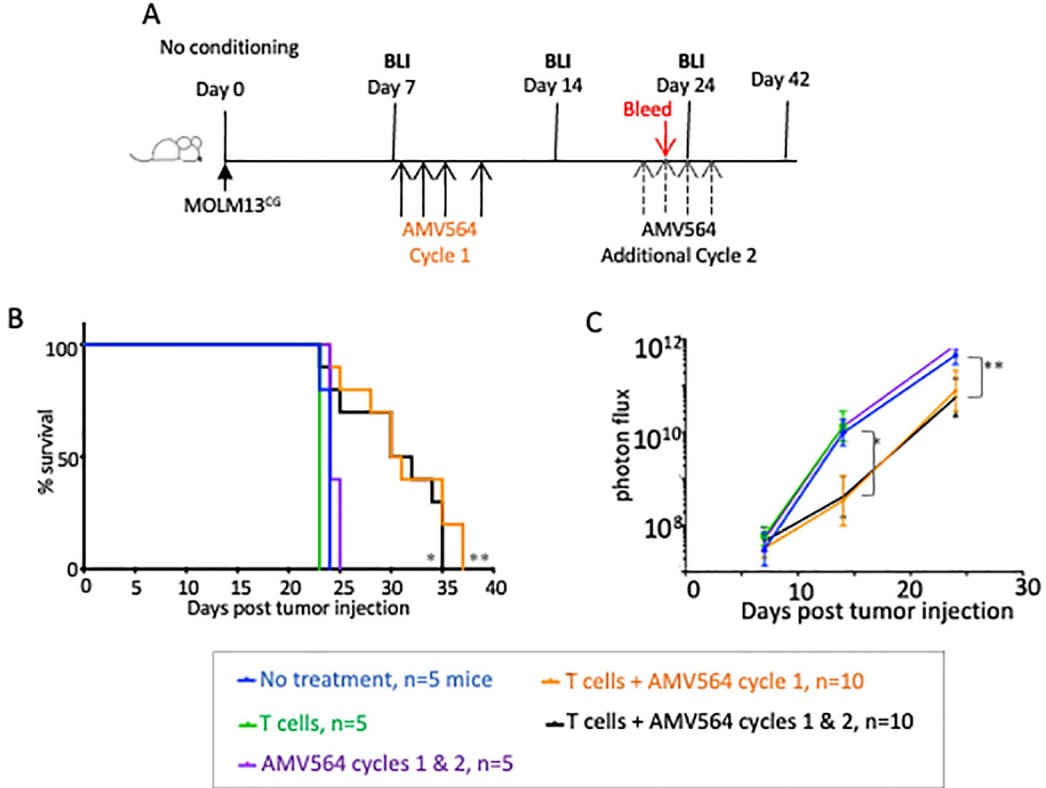

**Fig 2. Effect of treatment cycles on clearance of MOLM13$^{CG}$ cells engrafted in NSG mice.** (A) Schema. NSG mice were injected i.v. with 3.3 x 10$^3$ tumor cells seven days before injecting 2 x 10$^6$ purified human T cells. AMV564 (5 mcg) was administered to mice in a first cycle on days 7, 8, 9, and 11. One group of mice was then injected with an additional cycle of drug on days 22–25. (B) Mouse survival. *Untreated vs. T cells + both cycles of AMV564, p = 0.0213. **Untreated vs. T cells + single cycle AMV564, p = 0.005. (C) Kinetics of mean tumor expansion measured by serial BLI (shown as photon flux/second). *Untreated vs. T cells + AMV564 cycle 1 and untreated vs. T cells + AMV564 cycles 1 & 2 on day 14, p<0.0001. **Same groups on day 24, p≤0.0072. N = 1 experiment. N = 5 mice/group for controls and n = 10 mice/group for mice treated with both T cells and AMV564.

in at a rate in parallel with the control mice, albeit on a delayed time course. Considering that in the first drug treatment cycle the last injection of AMV564 was on day 11, then by the time of the first BLI on day 14, little drug remained to redirect T cells. We reasoned that at this point, either drug concentration or the relative E:T ratio might have been insufficient to control the tumor cells. Consequently, in the next experiment we changed two parameters. We increased the number of healthy donor T cells injected into all animals to 8 x 10$^6$ and also compared the use of both the original 5 mcg AMV564/injection and a higher dose of AMV564 (25 mcg) in this setting (Fig 3A). Compared to the previous experiment, median survival of tumor engrafted mice was significantly increased (Fig 3B). No mice in the control groups survived beyond day 28. The dose of AMV564 had a modest impact on survival (Fig 3C), but the additional T cells provided a substantial impact on tumor elimination even at the same dose as used in the previous experiment: 5mcg AMV564 (Fig 3C). In fact, based on BLI measurements, some mice in both the 5 mcg and 25 mcg groups completely cleared their tumors in combination with the allogeneic T cells (Fig 3D).

On day 21, there were very few tumor cells in the blood of any mouse that was administered allogeneic T cells, either with or without AMV564 (S3A Fig). However, BLI indicated high

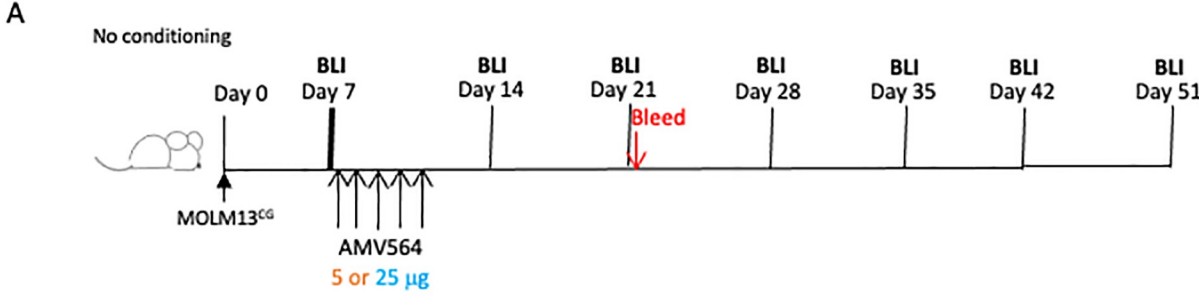

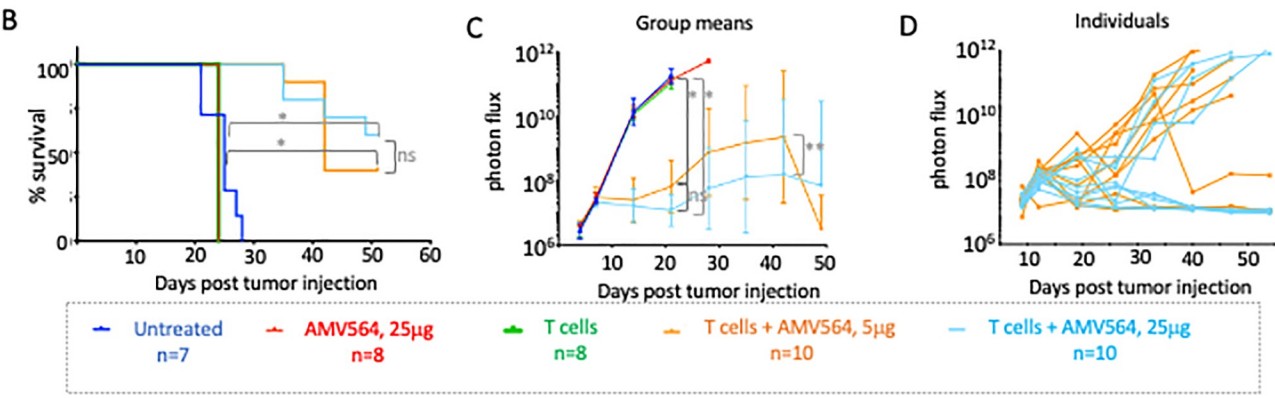

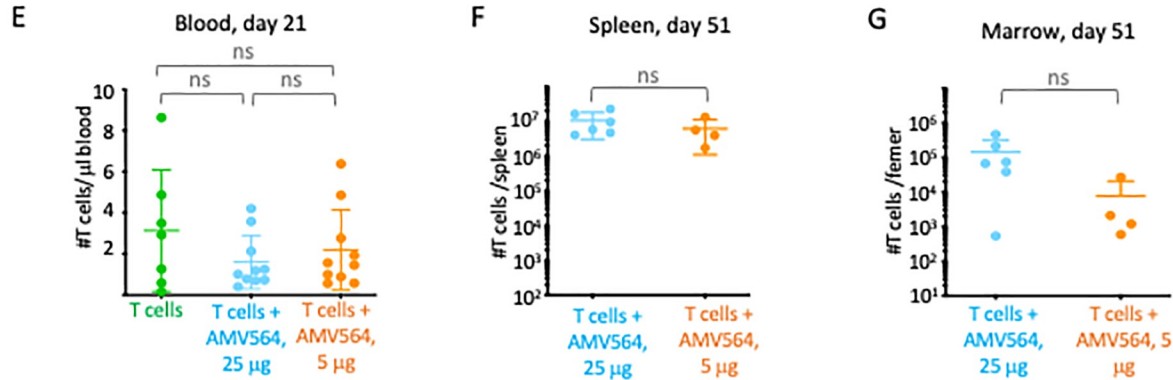

**Fig 3. Effect of T cell and AMV564 dose on clearance of MOLM13$^{CG}$.** (A) Schema. On day 7 post i.v. injection of 3.3 x 10$^3$ MOLM13$^{CG}$ cells, mice were injected with a single dose of 8 x 10$^6$ T cells, i.v., and a single 5-day cycle of AMV564 (5 or 25 mcg/day, i.v.). (B) Mouse survival. *Untreated vs. T cells + either 5 or 25 mcg AMV564/day, p<0.0001. ns = no significant difference. (C) Tumor expansion measured by serial BLI, mean group signal. *Untreated vs. T cells + either dose on day 21, p<0.0001. **Difference in slope between T cells + 5 mcg vs. T cells +25 mcg, p = 0.0136, however difference in values between doses on day 21 was not significant (ns). (D) Serial BLI signal in individual mice treated with T cells and either 5 or 25 mcg of AMV564. (E) Number of T cells/ul in the blood of all mice on day 21 or total number in the (F) bone marrow and (G) spleen of surviving mice on day 51. N = 1 experiment. N = 7 animals in untreated group. N = 8 animals in 25 mcg AMV group and in group treated only with T cells. N = 10 in groups treated with T cells in combination with either 5 or 25 mcg AMV564.

levels of MOLM13$^{CG}$ cells in the mice treated only with T cells. By day 24, that group of mice had died. By day 51, the mice treated with both drug and T cells remained clear of tumor cells in the spleen and bone marrow of all but one mouse (S3B and S3C Fig). This is explained by the correspondingly large numbers of T cells in these organs (Fig 3E–3G). This experiment led us to hypothesize that the dose of T cells injected was more critical for tumor clearance and survival of tumor engrafted mice than the dose of AMV564.

## Extramedullary tumor cells (derived from AML FAB M5 cell line) remain despite treatment with AMV564 + T cells

MOLM13 cells have a propensity to infiltrate extramedullary sites [10]. Consequently, we did a pathological examination of all treated animals surviving to or beyond day 42 in the previous experiment. In every mouse, even mice with no detectable BLI signal, there was histological evidence of residual extramedullary tumor in the skin, testes, epididymis, muscle of the small intestines, and/or unidentified soft tissue. This suggests poor penetration of either AMV564 or T cells or both into these extramedullary sites or that these sites promote an immunosuppressive effect on T cells, resulting in persistence of disease. Alternatively, there may have been antigen downregulation or escape (loss of CD33 expression) by these extramedullary tumors. Consequently, we evaluated CD33 expression on the MOLM13$^{CG}$ cells recovered from these extramedullary sites by flow cytometry. In no case was loss of antigen observed (S4 Fig). This suggests that rather than disease persistence due to antigen escape, AMV564 and/or the T cells may be limited in their penetration of these sites or that these sites are promoting T cell exhaustion or a direct immunosuppressive effect.

## AMV564 plus T cells is more effective when tumor burden is low

Since some mice in the previous experiment demonstrated complete loss of BLI-detectable tumors, we hypothesized that variation in disease burden at the time of treatment might influence effectiveness. Consequently, we compared the effect of initiating bispecific antibody treatment when tumor burden was low, as on day 3, or high, as on day 17 (Fig 4A). In both cases, healthy donor T cells were injected only on the first day of the drug cycle. We chose to use 5 mcg/day rather than 25 mcg/day since the experiment in Fig 3C showed that the number of T cells administered was more important than the AMV564 dose.

As predicted, injecting AMV564 and T cells when the tumor burden was low dramatically improved both survival and tumor clearance (Fig 4B and 4C). In fact, with early treatment, 7 out of 10 mice maintained baseline BLI signal throughout the observation period (S5 Fig). On day 12 (i.e., 5 days after the last injection of AMV564), T cell expansion had occurred only in the group of mice that had been injected with both T cells and AMV564 (Fig 4D). In other words, proliferation of T cells by this early time required that the T cells had been redirected by the AMV564. In so doing, the E:T ratio had shifted in favor of the T cells and, in fact, this resulted in clearance of the MOLM13$^{CG}$ cells from the blood (Fig 4E). In contrast, treating tumor-bearing mice when disease burden was high (day 17), and therefore the E:T ratio was quite low, failed to improve mouse survival (Fig 4B) or initiate a decrease in BLI signal by day 24 (Fig 4C). Nonetheless, it is noteworthy that by day 26, just two days later, there were high numbers of T cells in the blood (S6A Fig) and a substantially decreased number of tumor cells (S6B Fig). We hypothesize that AMV564 facilitated T cell mediated peripheral blood tumor clearance though with less effective tissue disease control, evidenced by persistent high BLI signal, resulting in death from extramedullary involvement.

Despite superior tumor suppression in the group of mice treated early, 4 of 10 died prior to day 60 (Fig 4B and S5 Fig). One of these mice with persistent tumor was sacrificed on day 32.

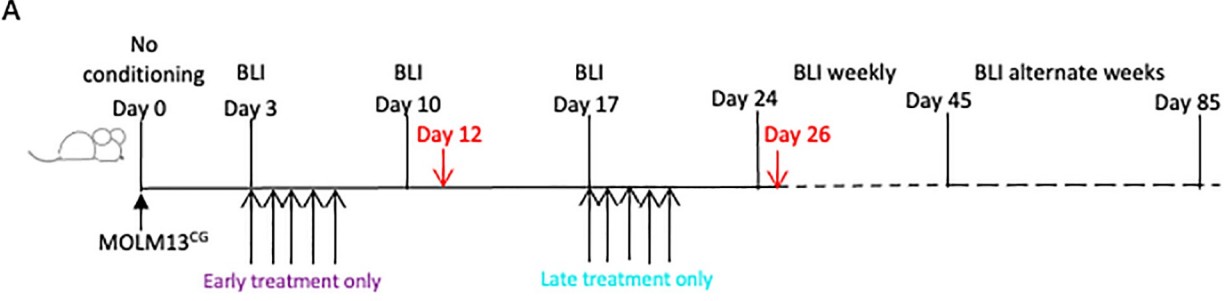

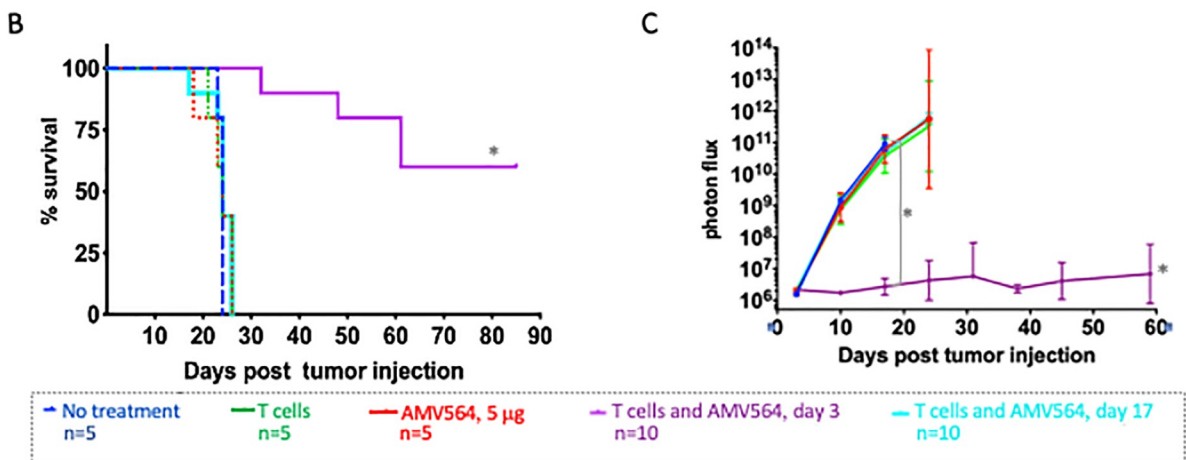

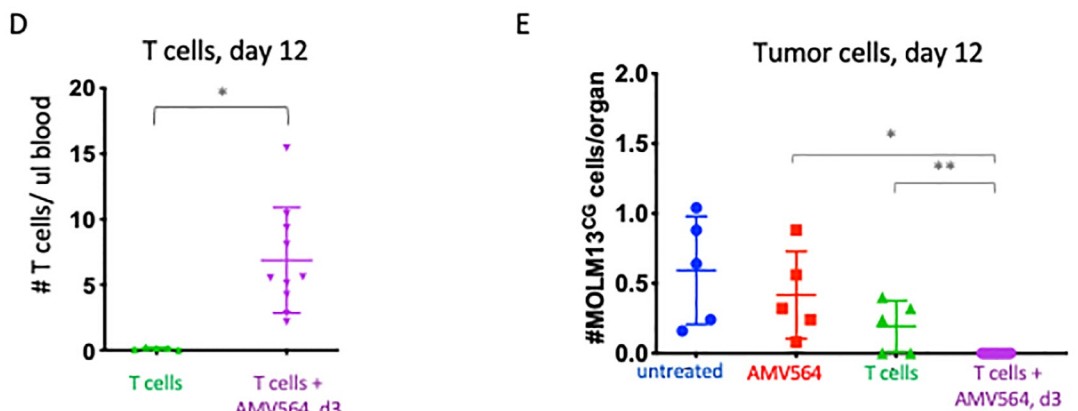

**Fig 4. AMV564 plus T cells is more effective at clearing tumor when treatment is initiated when tumor burden is low.** (A) Schema. Mice were injected i.v. with $3.3 \times 10^3$ MOLM13$^{CG}$. Mice were then injected i.v. with $8 \times 10^6$ T cells either on day 3 (early treatment) or 17 (late treatment) and a single 5 day cycle of 5 mcg AMV564/day was begun on the same day. Controls were injected only with AMV564 or only with T cells on day 3. (B) Mouse survival through day 85. *Untreated vs treatment beginning on day 3, p = 0.0001. (C) Serial BLI through day 59, group mean. * p<0.0001 for both the slope of untreated vs. early treatment begun on day 3 and for the comparison of BLI signal between the two on day 17. (D, E) Untreated controls and treated mice were bled on day 12. (D) T cells and (E) tumor cells in the blood on day 12. In (D), *p = 0.003. In (E), *p = 0.0005; ** p = 0.062. N = 1 experiment.

Two other mice initially responded but later "relapsed" at about day 37 with BLI-detectable tumors on their faces. These mice were sacrificed on day 62. Unexpectedly, the fourth mouse required sacrifice on day 48 due to morbidity (weight loss and hunching) despite having no BLI signal. This mouse also had the highest number of T cells in its blood on day 26 (55 T cells/mcL, S6A Fig).

### AMV564 can induce protective immunity *in vivo*

Given our observation in the experiment in Fig 3 that many mice had extramedullary disease, consistent with reports by others using MOLM13 cells [10], we wondered why only three mice relapsed in the previous experiment (S5 Fig). We speculated that in the other mice, persistence of infused allogeneic T cells, or perhaps even T cells responding to antigens on the tumor which they innately target, may be controlling disease long after AMV564 is cleared from the body.

To test whether T cells could still be detected in surviving mice on day 62 and whether they retain the ability to control proliferating tumor cells, blood was sampled from each of two mice that survived to day +62 in a new experiment (Fig 5A). Those same mice were then rechallenged with $3.3 \times 10^6$ MOLM13$^{CG}$ cells, the same number of tumor cells with which they were initially injected and which caused the never-treated control mice to die within 28 days. No additional treatment was provided before, during, or after rechallenge.

We had anticipated that the rechallenged mice might be able to suppress this large bolus of tumor cells somewhat, but expected that they would not survive more than a few days beyond the never-treated mice. Remarkably, despite receiving no additional treatment, both rechallenged mice survived ~53 more days post tumor injection, nearly twice as long as the never-treated controls (Fig 5B). In addition, and in contrast to the control group, the tumor was immediately suppressed in the rechallenged mice (Fig 5C), i.e., within just 3 days of injection. The BLI signal remained low throughout day 38, the last day the mice were imaged, thus providing evidence that the combined treatment with T cells and AMV564 had resulted in protective immunity.

As anticipated, residual T cells were detectable in the mice the day before they were rechallenged. The number of both CD4$^+$ and CD8$^+$ cells increased substantially over the ensuing 18 days (Fig 5D). The proportion of cells in each subset remained relatively similar over this time, and at all times the majority of T cells were effector and effector memory cells (Fig 5E). Notably, the overall increase in T cell numbers was primarily driven by a dramatic increase in the number of CD4$^+$ cells in all subsets (Fig 5F). The death of both rechallenged mice on day +R53 was again accompanied by symptoms consistent with a T cell-mediated xenogeneic GvHD-like disease (weight loss, scruffy hair, skin irritation).

### AMV564 activates autologous T cells in a primary AML xenograft to clear patient AML *in vivo* in NSG mice

In the previous experiments, AMV564 mediated an allogeneic T cell response to clear established tumor cells. We next sought to determine whether AMV564 can also induce a patient's own T cells to respond to and clear that same patient's primary CD33$^+$ AML *in vivo*. We chose a sample from a patient whose AML were CD33$^+$ and known to robustly engraft in sublethally irradiated NSG mouse bone marrow. This primary AML sample contained 98% tumor myeloblasts and only 2% T cells, i.e., an E:T ratio of 1:50, a lower ratio than was determined to be effective in our *in vitro* experiments. We reasoned that AMV564 treatment might still be successful *in vivo* at this seemingly low E:T.

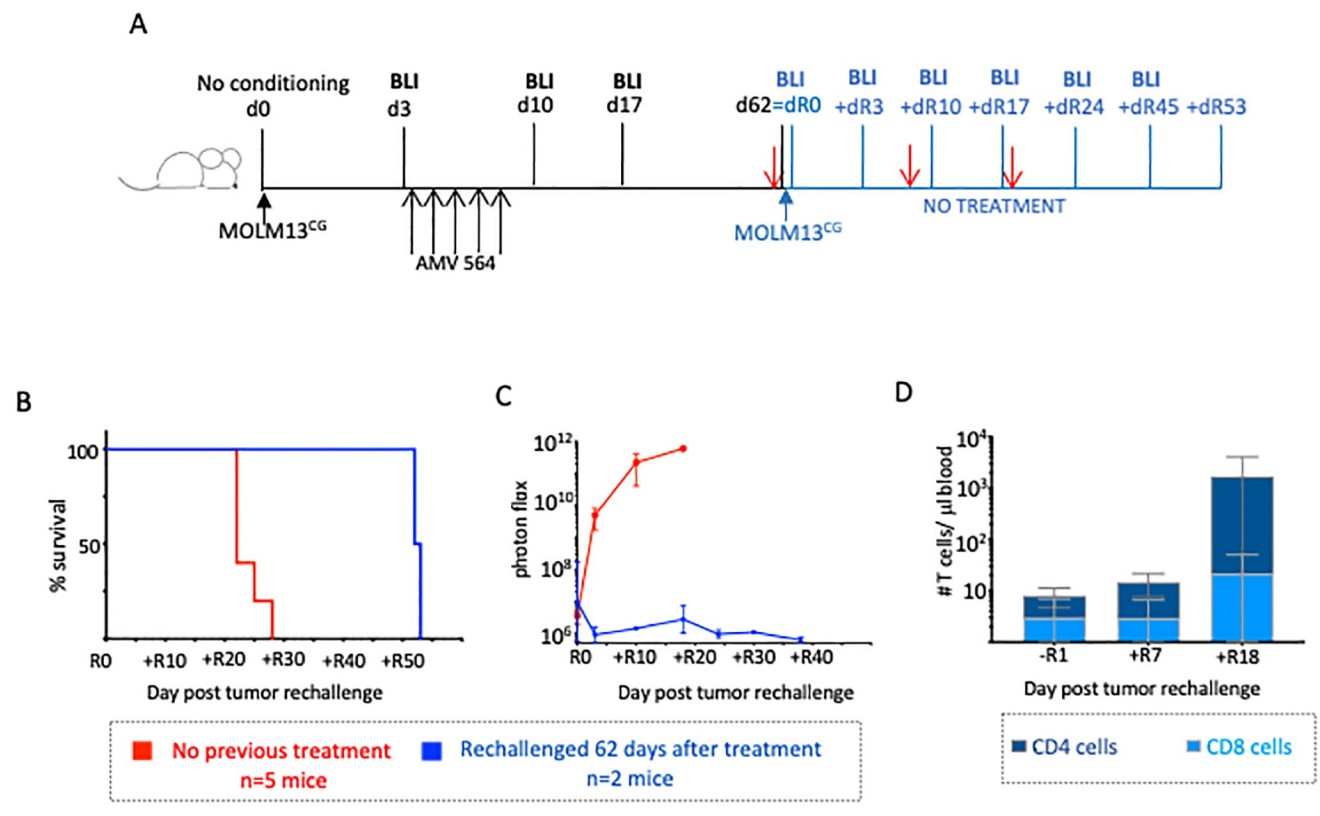

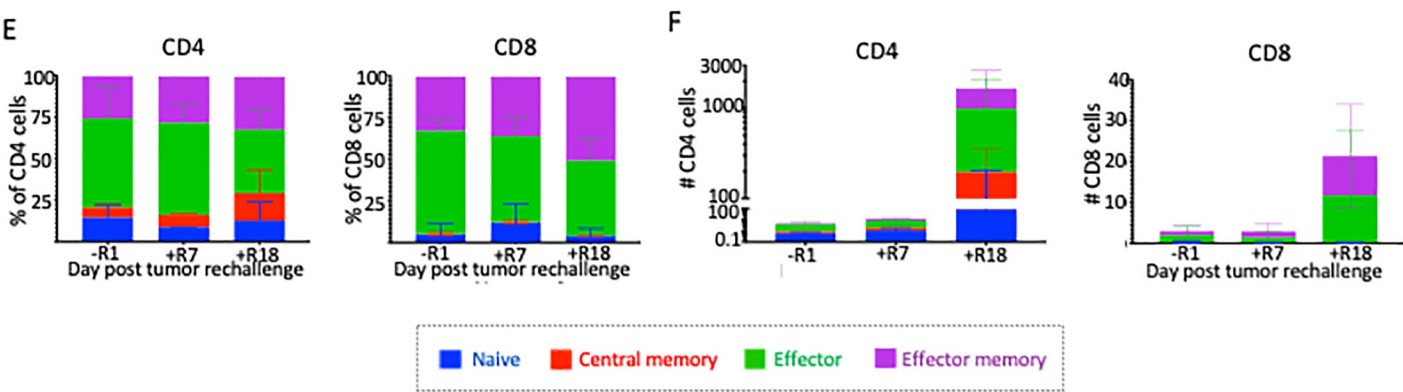

**Fig 5. Protective immunity mediated by persistent effector and memory T cells.** A. Schema. A group of five NSG mice were injected i.v. with $3.3 \times 10^3$ MOLM13$^{CG}$ cells. Three days later, they were treated with a single i.v. injection of $8 \times 10^6$ healthy donor T cells and 5 mcg AMV564 i.v. daily for 5 days. On day 62, the group of two surviving mice was rechallenged with $3.3 \times 10^3$ more MOLM13$^{CG}$ cells, but given no additional therapy (Day 62 = rechallenge day 0 [R0]). A group of five control mice was injected with tumor cells on the same day, and they also received no treatment. On days -R1 (one day before rechallenge), +R7, and +R18 (red arrows) mice were bled for flow cytometry. (B) Survival of rechallenged vs. control mice (never treated). (C) Mean BLI signal in rechallenged vs. control mice. (D) Increase in number of CD4$^+$ and CD8$^+$ T cells in the blood in rechallenged mice. *p = 0.0589. (E) Proportion and (F) number of CD4$^+$ and CD8$^+$ T cells in each subset in the blood before and after rechallenge. T cell subset phenotypic markers: naïve cells (CD197$^+$CD45RA$^+$), central memory cells (CD197$^+$CD45RA$^{neg}$), effector cells (CD197$^{neg}$CD45RA$^{neg}$), effector memory cells (CD197$^{neg}$CD45RA$^+$). N = 1 experiment.

We began AMV564 treatment within an hour of injecting the AML patient sample. Thirty-three days later, mice were sacrificed to assess tumor control in critical organs using flow cytometry (Fig 6A). In the absence of AMV564, AML blasts were detected in the bone marrow and spleens but not in the peripheral blood on day 33. However, after just 5 days of treatment, we could not detect AML blasts in the blood, bone marrow, or spleen of mice treated with either 5 or 50 mcg AMV564 (Fig 6B). The complete clearance of primary AML from all organs occurred despite no significant difference in the percentage or absolute numbers of CD4$^+$ and CD8$^+$ T cells in the bone marrow, spleen, and blood of human AML engrafted mice treated with T564 as compared with untreated mice (Fig 6C–6F).

## Discussion

While our study was small, it clearly demonstrated anti-tumor activity of AMV564, a novel bispecific TandAb against CD33$^+$ AML and AML cell lines *in vitro* and *in vivo*. Based on our studies, we hypothesize that early in treatment with the combination of AMV564 and T cells, the bispecific redirects, activates, and expands T cells in a TCR-independent manner, as expected, and thus reduces the tumor burden. Once the concentration of bispecific becomes negligible due to its half-life of ~18–23 h in mice [10], redirection is lost. At this point, control of any remaining tumor cells likely becomes reliant not only on the E:T ratio, but also on TCR-dependent mechanisms intrinsic to the T cells that were activated and expanded in the time window provided during the redirection. This could be either an allogeneic response or an intrinsic response to antigens on the MOLM13$^{CG}$ cells, e.g., CD33 or some other antigen, in a process akin to epitope spreading. Subsequent tumor control seems to require that the *in vivo* E:T ratio is adequately high when the drug concentration becomes negligible. This was likely not the case in the experiment described in Fig 2, since tumor proliferation resumed once the drug was depleted. However, in the case of later experiments, a larger number of T cells was injected, became activated and proliferative during the brief drug treatment, and in at least some of the mice, the threshold E:T ratio was then likely surpassed when the drug concentration became neglible. However when AMV564 is administered too late after AML infusion in NSG mice, it still reduces tumor burden (S6 Fig), but can also induce a highly inflammatory state, perhaps due to tumor lysis and/or to a GvHD-like disease mediated by the high concentration of T cells in the blood.

It is significant that this off-the shelf treatment can generate protective immunity, as seen in the rechallenge experiment (Fig 5). Here, a short 5-day treatment resulted in the survival of two out of five mice for 62 days, and in both of these mice, donor effector T cells and effector memory T cells persisted. Within just 3 days of rechallenge, both mice completely eliminated AML cell line proliferation as measured by BLI, without any additional AMV564. Our studies raise the question of exactly how the T cells in the rechallenged mice recognized and eliminated MOLM13$^{CG}$ cells without additional AMV564. Since no effort was made to match the HLA haplotypes of the donor T cells and MOLM13$^{CG}$ cells, the specific donor T cells used in Fig 5 possessed some alloreactivity towards the MOLM13$^{CG}$ cells likely resulting in prolonged tumor control. However when the same number of normal T cells from multiple different donors (all of which were HLA-mismatched with the MOLM13$^{CG}$ cells) were infused into NSG mice engrafted with MOLM13$^{CG}$ cells, they were uniformly unable to reduce *in vivo* proliferation of MOLM13$^{CG}$ cells as demonstrated by BLI or to delay time to death from progressive leukemia (Figs 3B–3D and 4B and 4C). Consequently. the brief redirection by the bispecific was critical for jump-starting a tumor-suppressive immune response.

We recently reported that a related bispecific dual affinity retargeting agent (DART) targeting CD123, flotetuzumab, in addition to mediating tumor cytotoxicity, could directly induce

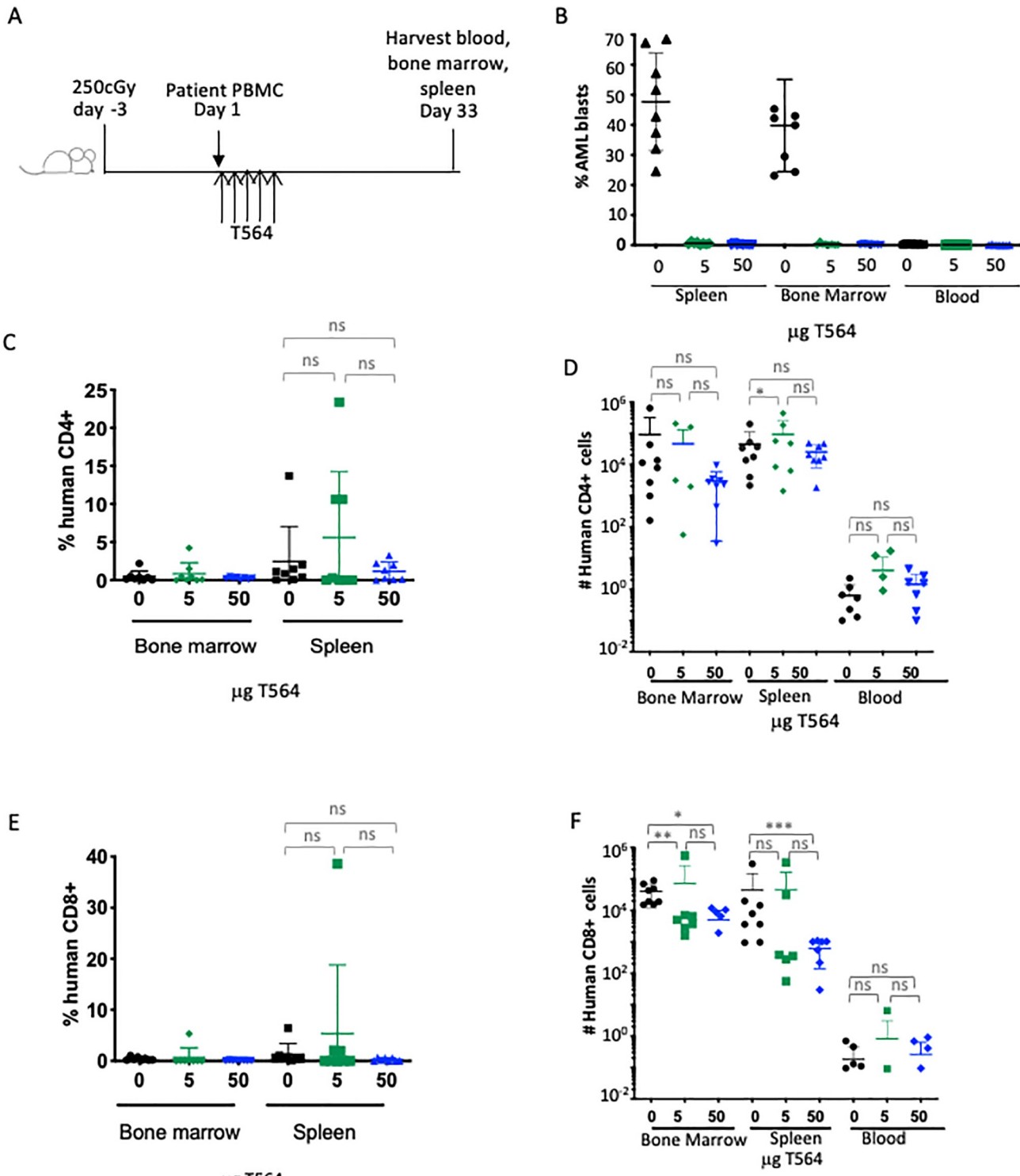

**Fig 6. AMV564 induces patient T cells to clear autologous tumor cells in NSG mice.** (A) Schema. PBMC (5 x 10[6]) from a CD33+ AML patient sample containing only 2% CD3+ T cells and 98% CD33+ myeloblasts were injected i.v. into NSG mice irradiated (250cGy) 3 days earlier. One hour later and once each day for a total of 5 days, TandAb AMV564 (0, 5, or 50 mcg; n = 8 mice/group) was injected i.v. Mice were euthanized on day 33, the harvested organs were analyzed by flow cytometry to detect human T cells and human AML using 7AAD, human CD34, CD45, CD123, CD3, and CD4 immunophenotyping. Shown are the percentage of cells harvested from the blood that were (B) AML blasts, and (C-F) the percentage and absolute number of CD4+ and CD8+ T cells. In F, * p = 0.0350, **p = 0.0461, ***p = 0.0095. N = 1 experiment. N = 8 mice/group.

the upregulation of MHC class II in AML cell lines and in primary AML [11]. Although bispecifics like AMV564 (which targets CD33 on AML blasts) together with T cells can directly mediate killing of AML via a MHC independent mechanism, activation of T cells by bispecifics after target engagement can also enhance MHC-dependent killing of AML cells via local release of interferon gamma and subsequent upregulation of MHC class II on AML blasts [12]. This enhanced MHC-dependent allogeneic response may also be responsible for the persistence of donor T cells after treatment with AMV564 in MOLM13[CG]-engrafted mice, and for the lack of tumor progression coupled with rapid expansion of donor T cells in mice after tumor rechallenge.

The fact that memory T cells persisted until at least day 62 (Fig 5), is possibly due to intermittent low-level exposure of the T cells to recalcitrant extramedullary tumors that provide a renewable source of antigen stimulation. We detected these in the skin, testes, epididymis, intestines, and other unidentified soft tissues even when mice appeared to be healthy and BLI signal was undetectable (Fig 3). We hypothesize that the memory T cells keep those tumors at bay, and account for the rapid anamnestic response that cleared MOLM13[CG] cells within 3 days of a rechallenge (Fig 5).

It is unclear whether the protective immunity results from persistence of innately anti-AML T cells from the donor, or simply persistence of alloreactive T cells that can both eliminate MOLM13[CG] cells and result in graft vs. host disease. Of note is that we show that when T cells and AML blasts are derived from the same patient, the complete elimination of AML *in vivo* can also be achieved in the absence of any potential alloreactivity, and even in the setting of extremely low E:T ratios. These findings warrant further investigation in an autologous setting and in various other model systems into how much tumor reduction has occured immediately after the brief drug treatment; the ratio of E:T in the blood, bone marrow, and spleen within 24h of the last dose; and whether these redirected T cells would respond *in vitro* to tumor cells both with and without the CD33 antigen to which the bispecific was directed.

There are important distinctions in the design and outcome of our study and those using other bispecifics targeting CD33 and CD3. In our study, AMV564 was able to promote prolonged tumor control for over 12 weeks in mice after a single brief (5-day) treatment cycle. A single injection of non-activated T cells without any supplemental supportive cytokines was needed. Studies using other bispecifics used daily treatments for longer periods [13, 14], more or activated T cells or cytokines [15], or were not designed to assess mouse survival [14, 16]. No control bispecific agents or activated T-cells were utilized in the animal work, which remains a limitation. Our experiments show that either by study design or attributes of this particular bispecific, it is not only possible for bispecifics directed against AML to have strong efficacy in a preclinical setting specifically using NSG recipient mice, but also for them to induce protective immunity using a short uncomplicated treatment regimen.

To our knowledge, this is the first report of a bispecific that induces protective immunity in an AML mouse model, although this has been reported in two other cancer models [17, 18]. Since these reports used syngeneic tumors, the potential alloreactivity seen in our studies may not be the only possible mediator of protection from tumor rechallenge. At this point, it is unclear what additional factors promote protective immunity, although certainly reagent design, therapeutic schedules, and the overall disease burden should be considered. Additional investigation into the mechanisms for generating protective immunity using this and other bispecifics are clearly warranted to optimize potential patient benefits while minimizing potential harm. These data establish AMV564 as an effective CD33 x CD3 TandAb bispecific that may provide benefit to patients with relapsed or refractory AML when explored in future clinical trials.

## Supporting information

**S1 Fig. Kinetics of tumor expansion as measured by BLI in mice injected with T cells only, or T cells plus one or two cycles of AMV564.** Same data as in Fig 2C, but rather than displaying data as group mean, the BLI signal in individual mice is shown. N = 5 mice/group for controls and n = 10 mice/group for mice treated with both T cells and AMV564.
(TIFF)

**S2 Fig. Tumor expansion as measured by flow cytometry in mice injected with T cells only, or T cells plus one or two cycles of AMV564.** Same experiment as in Fig 2. On day 23 all mice were bled and their MOLM13$^{CG}$ burden was determined. Flow cytometry markers = GFP, muCD45, huCD45, huCD33, huCD3, and Live/Dead stain. *p = 0003. **, p = 0.0164. N = 5 mice/group for controls and n = 10 mice/group for mice treated with both T cells and AMV564.
(TIFF)

**S3 Fig. Effect of AMV564 dose on tumor cell numbers in various organs.** Same experiment as Fig 3. Number of tumor cells in the (A) blood on day 21 or the (B) the bone marrow and (C) spleen on day 26. In (A), *p < 0.001, ** p = 0.0003. Flow markers = GFP, muCD45, huCD45, huCD33, huCD3, huCD4, huCD8, and 7-AAD. N = 7 animals in untreated group. N = 8 animals in 25 mcg AMV group and in group treated only with T cells. N = 10 in groups treated with T cells in combination with either 5 or 25 mcg AMV564.
(TIFF)

**S4 Fig. CD33 expression on MOLM13$^{CG}$ cells prior to injection (left) and harversted from extramedullary site.** Same experiment as described in main text Fig 3. Representative data from one sample. N = 7 animals in untreated group. N = 8 animals in 25 mcg AMV group and in group treated only with T cells. N = 10 in groups treated with T cells in combination with either 5 or 25 mcg AMV564.
(TIFF)

**S5 Fig. Serial BLI in individual mice whose group mean BLI is shown in Fig 4C.** In the early treatment group, one mouse had no BLI signal, but required sacrifice on day 48 due to GvHD-like symptoms. For mice left untreated, treated only with T cells or only with AMV564, N = 5 mice/ group. For mice treated with both T cells and AMV564, N = 10 mice/group.
(TIFF)

**S6 Fig. On day 26 the number of T cells in the blood was greater and the number of tumor cells in the blood was lower in both the early and late AMV564 + T cells groups relative to only T cells.** In A, *p = 0.0039, ** = 0.0005. In B, *p = 0.0001, **p<0.0001, ***p = 0.0005. For mice left untreated, treated only with T cells or only with AMV564, N = 5 mice/ group. For mice treated with both T cells and AMV564, N = 10 mice/group.
(TIFF)

**S1 Table. Morbundity assessment of animals.** To assess moribundity, points are assigned to each of the 5 criteria below. A total score of 6 or higher or 30% weight loss, regardless of score, requires euthanasia of the animal.
(DOCX)

**S2 Table. Details of animal usage in individual experiments.**
(DOCX)

**S3 Table. Antibodies used for flow cytometry.**
(XLSX)

**S1 Checklist. ARRIVE guidelines 2.0: Author checklist.**
(PDF)

## Acknowledgments

The authors thank Dr. Joel C. Eissenberg for reading and editing the manuscript.

## Author Contributions

**Conceptualization:** Linda G. Eissenberg, Michael P. Rettig, Victoria Smith, Tae H. Han, John F. DiPersio.

**Data curation:** Tae H. Han.

**Formal analysis:** Julie K. Ritchey, Michael P. Rettig, Kiran Vij, Feng Gao.

**Funding acquisition:** Linda G. Eissenberg, John F. DiPersio.

**Investigation:** Julie K. Ritchey, Dilan A. Patel.

**Methodology:** Julie K. Ritchey.

**Project administration:** Dilan A. Patel.

**Supervision:** Victoria Smith, Tae H. Han, John F. DiPersio.

**Validation:** John F. DiPersio.

**Visualization:** Dilan A. Patel, John F. DiPersio.

**Writing – original draft:** Linda G. Eissenberg.

**Writing – review & editing:** Linda G. Eissenberg.

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
