## [Decision Letter · Decision Letter 0]

16 Feb 2024

PONE-D-23-41952Control of acute myeloid leukemia and generation of immune memory in vivo using AMV564, a bivalent bispecific CD33 x CD3 T cell engagerPLOS ONE

Dear Dr. Patel,

Thank you for submitting your manuscript to PLOS ONE. After careful consideration, we feel that it has merit but does not fully meet PLOS ONE’s publication criteria as it currently stands. Therefore, we invite you to submit a revised version of the manuscript that addresses the points raised during the review process.

We look forward to receiving your revised manuscript.

Kind regards,

Anilkumar Gopalakrishnapillai

Academic Editor

PLOS ONE

Journal Requirements:

"The work was supported by National Institutes of Health (NIH)/National Cancer Institute (NCI): R35 CA210084 (JFD), NIH/NCI: P50 CA171963 NCI (JFD) and NIH/NCI R50 CA211466-06 (MPR)."

4. Thank you for stating the following in your Competing Interests section: "No"

6. Please upload a copy of Supporting Information Table. S1 - 23 Tables and S1 Checklist. which you refer to in your text on page 24.

7. We notice that your supplementary figures are uploaded with the file type 'Figure'. Please amend the file type to 'Supporting Information'. Please ensure that each Supporting Information file has a legend listed in the manuscript after the references list.

Reviewers' comments:

Reviewer's Responses to Questions

**Comments to the Author**

1. Is the manuscript technically sound, and do the data support the conclusions?

Reviewer #1: Yes

Reviewer #2: Yes

2. Has the statistical analysis been performed appropriately and rigorously? 

Reviewer #1: Yes

Reviewer #2: Yes

3. Have the authors made all data underlying the findings in their manuscript fully available?

Reviewer #1: Yes

Reviewer #2: No

4. Is the manuscript presented in an intelligible fashion and written in standard English?

Reviewer #1: Yes

Reviewer #2: Yes

5. Review Comments to the Author

Reviewer #1: In this manuscript Eissenberg et al describe bispecific T engager AVM564 as a potential therapeutic for AML. This is a well written manuscript that describes additional preclinical work with a novel agent with potential for translation to the clinic and intriguing data suggesting potential lasting immunologic effects. In this resubmission the authors address the prior reviewers critiques adequately and clearly describe the limitations around pursuing additional experiments for this publication.

Critiques are primarily limited to grammatical edits and suggestions to improve clarity for the reader.

Minor Comments:

Spelling error throughout: Albumin rather than Albumen

Would move the FAB subtype comment to the Cells subheading

Would clearly state the limitation that no control bispecific or activated T cells were utilized in the animal work.

Line 353 extra space in initially

Reviewer #2: The manuscript by Eissenberg et al reports the preclinical efficacy of bispecific antibodies targeting CD33. That antibody treatment can impart tumor immunity demonstrated by protection against rechallenge is an interesting observation. In addition to using AML cell lines, data in a primary AML sample with 2% T cells are presented.

Major

No survival data is reported for the primary patient sample study in mice. It would be interesting to see if rechallenge with primary sample in this model is ineffective and does not lead to leukemia.

Did not find Figs. 4F and 4G. Although BLI signal was high in these mice, the authors claim that the mice did not die from tumor, referring to Fig. 4G. This is confusing.

The following minor point should be addressed.

Consider revising “induces protective immunity in an immunodeficient model’ in the abstract, as it is confusing.

Redirect CD3+ cells to CD3 epsilon? First sentence in introduction, please revise.

What would be the source of allogeneic T cells in a patient? Will immunity be restricted to patients who have undergone HCT?

Please correct typo – albumin instead of albumen

Can the authors comment on why 4 doses were used in Fig 2, while 5 doses were used in other experiments

6. PLOS authors have the option to publish the peer review history of their article (what does this mean?). If published, this will include your full peer review and any attached files.

Reviewer #1: No

Reviewer #2: **Yes: **Sonali Barwe

---

## [Author Response · Author response to Decision Letter 0]

20 Feb 2024

Reviewer #1: In this manuscript Eissenberg et al describe bispecific T engager AVM564 as a potential therapeutic for AML. This is a well written manuscript that describes additional preclinical work with a novel agent with potential for translation to the clinic and intriguing data suggesting potential lasting immunologic effects. In this resubmission the authors address the prior reviewers’ critiques adequately and clearly describe the limitations around pursuing additional experiments for this publication.

Critiques are primarily limited to grammatical edits and suggestions to improve clarity for the reader.

Author response: Thank you for taking the time to review our resubmission and for the constructive comments to improve the content and formatting. 

Minor Comments:

Spelling error throughout: Albumin rather than Albumen

Would move the FAB subtype comment to the Cells subheading

Would clearly state the limitation that no control bispecific or activated T cells were utilized in the animal work.

Line 353 extra space in initially. 

Author response: Thank you for reviewing and identifying these errors. We have corrected the spelling of “albumin” in multiple areas. The FAB subtype was moved to the “Cells” subheading rather than the first line of the paragraph. The limitation that no control bispecific or activated T cells were utilized in the animal work was added (lines 516-518). The extra space in the word “initially” was removed. 

Reviewer #2: The manuscript by Eissenberg et al reports the preclinical efficacy of bispecific antibodies targeting CD33. That antibody treatment can impart tumor immunity demonstrated by protection against rechallenge is an interesting observation. In addition to using AML cell lines, data in a primary AML sample with 2% T cells are presented.

Major

No survival data is reported for the primary patient sample study in mice. It would be interesting to see if rechallenge with primary sample in this model is ineffective and does not lead to leukemia.

Author response: Thank you for bringing this to our attention. We agree that this is worthy of rigorous repeat investigation though are not able to perform the rechallenge experiment due to lack of reagent issued from the company. 

Did not find Figs. 4F and 4G. Although BLI signal was high in these mice, the authors claim that the mice did not die from tumor, referring to Fig. 4G. This is confusing.

Author response: We appreciate the reviewers bringing up this discrepancy. The figure has now been correctly labeled as Fig 6S A-B rather than Fig 4F-G. The hypothesized mechanism of death has been edited for clarity (lines 354-357). 

The following minor point should be addressed.

Consider revising “induces protective immunity in an immunodeficient model” in the abstract, as it is confusing.

Author response: Thank you for pointing out the lack of clarity. The sentence has been changed to “facilitates memory responses” (line 19). 

Redirect CD3+ cells to CD3 epsilon? First sentence in introduction, please revise. 

Author response: We appreciate the feedback on suboptimal sentence phrasing and have edited for clarity that “…facilitate immune synapse between CD3 epsilon on T cells and selected differentially expressed antigens on tumors” (lines 41-42). 

What would be the source of allogeneic T cells in a patient? Will immunity be restricted to patients who have undergone HCT?

Author response: The source of T cells was healthy human donors. Anti-tumor efficacy correlates with T cell dose. We anticipate higher efficacy with T cells following HCT though tumor response with other forms of administration also. 

Please correct typo albumin instead of albumen

Author response: “Albumen” has been corrected to “albumin” throughout the manuscript. 

Can the authors comment on why 4 doses were used in Fig 2, while 5 doses were used in other experiments?

Author response: Thank you for this question. An additional dose was added based on the favorable responses without toxicity noted with 4 doses.

---

## [Editor Report · Decision Letter 1]

23 Feb 2024

Control of acute myeloid leukemia and generation of immune memory in vivo using AMV564, a bivalent bispecific CD33 x CD3 T cell engager

PONE-D-23-41952R1

Dear Dr. Patel

We’re pleased to inform you that your manuscript has been judged scientifically suitable for publication and will be formally accepted for publication once it meets all outstanding technical requirements.

Kind regards,

Anilkumar Gopalakrishnapillai

Academic Editor

PLOS ONE
---

## [Editor Report · Acceptance letter]

7 Mar 2024

PONE-D-23-41952R1 

PLOS ONE

Dear Dr. Patel, 

I'm pleased to inform you that your manuscript has been deemed suitable for publication in PLOS ONE. Congratulations! Your manuscript is now being handed over to our production team.

Kind regards, 

on behalf of

Dr. Anilkumar Gopalakrishnapillai 

Academic Editor

PLOS ONE